# Grapevine Leafroll-Associated Virus 3 in Single and Mixed Infections Triggers Changes in the Oxidative Balance of Four Grapevine Varieties

**DOI:** 10.3390/ijms24010008

**Published:** 2022-12-20

**Authors:** Katarina Hančević, Mate Čarija, Sandra Radić Brkanac, Emanuel Gaši, Matevž Likar, Goran Zdunić, Marjana Regvar, Tomislav Radić

**Affiliations:** 1Institute for Adriatic Crops and Karst Reclamation, Put Duilova 11, 21000 Split, Croatia; 2Faculty of Science, University of Zagreb, Rooseveltov trg 6, 10000 Zagreb, Croatia; 3Biotechnical Faculty, University of Ljubljana, Večna pot 111, 1000 Ljubljana, Slovenia

**Keywords:** GLRaV-3, grapevine, interactions, oxidative stress, physiological response, symptoms

## Abstract

With the aim to characterize changes caused by grapevine leafroll-associated virus 3 (GLRaV-3) singly or in coinfection with other viruses and to potentially determine genotype-specific or common markers of viral infection, thirty-six parameters, including nutrient status, oxidative stress parameters, and primary metabolism as well as symptoms incidence were investigated in ‘Cabernet Franc,’ ‘Merlot,’ ‘Pinot Noir,’ and ‘Tribidrag’ grapevine varieties. Host responses were characterized by changes in cellular redox state rather than disturbances in nutrient status and primary metabolic processes. Superoxide dismutase, hydrogen peroxide, and proteins were drastically affected regardless of the type of isolate, the host, and the duration of the infection, so they present cellular markers of viral infection. No clear biological pattern could be ascertained for each of the GLRaV-3 genotypes. There is a need to provide a greater understanding of virus epidemiology in viticulture due to the increasing natural disasters and climate change to provide for global food production security. Finding grape varieties that will be able to cope with those changes can aid in this task. Among the studied grapevine varieties, autochthonous ‘Tribidrag’ seems to be more tolerant to symptoms development despite numerous physiological changes caused by viruses.

## 1. Introduction

The biology of GLRaV-3, a globally distributed *Ampelovirus* that infects *Vitis vinifera* L. plants, is still an enigma. Its written history begins with a description of symptoms in European grapevine plants, some of them herbarium archived at the end of the 19th century [1]. Since that indirect confirmation of viral disease presence, we have come to more than 40 fully sequenced isolates of GLRaV-3. As a single-stranded positive RNA virus, viral RNA-dependent RNA polymerase contributes to genetic variability leading to the accumulation of genetically diverse variants, often called “quasi-species” [2]. Based on the phylogenetic analyses, GLRaV-3 variants are classified into eight basic phylogenetic groups: I, II, III, V, VI, VII, IX, and X [3], with those belonging to phylogenetic groups I and II, globally the most represented. With more samples analyzed, more variants are continuously being discovered [4], pushing the need to systematize viral genome regions.

As a member of the family Closteroviridae [5], whose world-well-known representative, Citrus tristeza virus genome and biological properties are deeply characterized and those parameters correlated [6,7], the understanding of GLRaV-3 pathobiology in general and the connection between different GLRaV-3 isolates and genotypes to biological symptoms is far from complete [8,9]. This is not surprising, considering that grapevine response to viral infection is not of generic nature but almost elusive, as the biotic and abiotic factors affecting the outcome are numerous, from virus and plant genotype to inoculation procedure, experimental period, the time of sampling, as well as the environmental and growth conditions. Quantification of different GLRaV-3 genomic variants in white and red cultivars implies that variant amplification is variety-dependent [10,11,12]. The study of transmission efficiency revealed that variants are biologically distinct in terms of vector transmissibility [13] which contributes to the presumption that genomic variants have different biological properties. However, whether different GLRaV-3 isolates, either singly or in coinfection with other viruses and their genomic variants, differ in their ability to induce distinct changes in their hosts, and if so, to which extent they are unique in provoking host reactions, is still unknown.

Regardless of virus variants, the host response to systemic viral infection includes the slow accumulation of reactive oxygen species (ROS), allowing systemic virus spread and generating the state of oxidative stress [14]. In so-called compatible plant-pathogen reactions, no resistance (R) genes are engaged in the process after virus infection, leaving the plant with no native immune response to fight viral attacks. As a result, visible symptoms in the host plants appear [15]. The absence of R gene expression undoubtedly contributed to the long and almost mutualistic coexistence of GLRaV-3 with its grapevine hosts but also directed the grapevine response to virus attack.

Disturbed sugar and anthocyanin metabolism [16] in the host grapevine are two major events following GLRaV-3 infection. Massive accumulation of these biological compounds occurs in leaves, thus, provoking other vital processes to be disrupted, such as photosynthesis [15,17,18]. At the same time, berries lack sugar and anthocyanin, leaving the yield immature and not enough sweet as well as insufficient pigmented [19]. The background of these processes in the prism of compatible pathogen reactions is difficult to explain, but different approaches in terms of gene expression and physiological and biochemical analysis add value to this issue [19,20,21,22].

Considering family affiliation to *Closteroviridae*, where genomic variants may show different biological patterns, differences in transmission efficiency among GLRaV-3 genotypes, and variability in the concentration of GLRaV-3 genomic variants, we can assume a specific variant-host interaction pattern that potentially could be evidenced via unique pathobiology.

With that presumption, physiological, morphological, and biochemical plant changes were investigated to elucidate *Vitis vinifera* response to different virus isolates, which necessarily contained GLRaV-3 singly or in coinfections with other economically important viruses. In the three-year experiment conducted in the controlled environment of a greenhouse, 36 different parameters potentially contributing to a specific virus-induced pathobiological profile were studied in four different grapevine varieties to understand plant–virus interactions and their consequences better.

## 2. Results

### 2.1. Laboratory Virus Detection

Enzyme-linked immunosorbent assays (ELISA) test was not a reliable test for virus detection in plants harboring such short-term infection. Three months after virus transmission, only 10 positive plants out of 260 inoculated were detected, and six months after transmission, only 16 were positive. All plants that tested negative on the ELISA test were confirmed to be GLRaV-3 infected by standard reverse transcription-polymerase chain reaction (RT-PCR) or nested PCR.

### 2.2. Biochemical and Physiological Measurements in 2020 and 2021

#### 2.2.1. Leafroll-Induced Changes

A permutational analysis of variance (PERMANOVA) out of the measured dataset (Appendix A) indicated that grape variety (R^2^ = 0.16, *p* = 0.001) and leafroll (LR) treatment (R^2^ = 0.02, *p* = 0.02) both had significant effects on the measured morphological and physiological parameters. Furthermore, we observed a statistically significant interaction between both (R^2^ = 0.09, *p* = 0.002). To clarify the contribution of individual measured parameters, we performed a discriminant analysis of principal components (DAPC).

DAPC analysis for grapevines infected with LR isolates showed less separation in the first year (Figure 1a). Only ‘Merlot’ was clearly separated by LD1, which showed significant loadings of a parameter related to color change (avg h; Figure 1b). LD2 separated ‘Pinot Noir’ from the remaining varieties, t.i. ‘Cabernet Franc’ and ‘Tribidrag,’ as well as controls from the LR-treated plants with the exception of the ‘Tribidrag’ variety. LD2 showed significant loadings of internode length (Figure 1c). In the second year, avg h remained an important factor showing significant loadings for LD2 (Figure 1f). In contrast, internode length was replaced by leaf Ca concentrations (Figure 1e). The separation between grapevine varieties in the second year was better, as ‘Cabernet Franc’ and ‘Tribidrag’ were also separated. The separation between control and virus-treated plants was also observed, with the exception of ‘Cabernet Franc.’ The spread of plants inoculated with different LR isolates was not increased from year to year.

Based on the results of the DAPC, we performed a Procrustes analysis on the variables that showed high loadings with LDs for the ordinations of both years, but overall, dissimilarity among the varieties increased from 2020 to 2021. Procrustes analysis revealed that the variety and virus treatments were significantly correlated in both years. ‘Tribidrag’ variety showed the biggest difference between years (Figure 2a). From the first to the second year, ‘Tribidrag’ and ‘Cabernet Franc’ showed changes in different directions, as DAPC also showed better separation of both varieties during the second year. ‘Merlot’ showed the smallest changes between the years for the selected parameters.

If we consider all parameters tested in relation to the control plants of every variety (Figure 3a), stress parameters were the most affected in the first and second years. The trend of causing stable changes over the years refers mostly to superoxide dismutase (SOD), peroxide, malondialdehyde (MDA), and proteins. Accordingly, the macroelement nitrogen was changed in all indicators, and the trend of its increasing significance was observed with the progression of the infection. On the contrary, the significance of Fe diminished with time. Changes in the concentration of carotenoids, chlorophyll *a* and *b* also point to an important role in the infection-related processes (Figure 3a).

#### 2.2.2. LR Genotype-Related Changes

To observe the potential specificity related to the LR genotype and yearly trend, we summarized all changed parameters in relation to the control plants for the LR genotype effect (Appendix A). Out of all LR genotypes analyzed, none could be firmly associated with any of the parameters tested, but the most severe changes were observed in SOD, peroxide, and proteins, respectively (Figure 4).

‘Merlot’ was the variety most affected by different genotypes (71; Appendix A) in 2020, but in 2021 affected parameters were reduced to a minimum. This projection was also shown by observing parameters presented in Figure 4, where ‘Merlot’ was the least affected by LR isolates compared to other varieties tested. As opposed to the second measurement, where significant genotype effects were more focused on stress parameters, in the first year, many other parameters of nutrient status (Fe, Mn, Zn, Mg) and primary metabolism (relative water content and membrane permeability) were disturbed. The parameters that were affected by the most genotypes and in all varieties in 2020 were Fe, SOD, peroxide, and proteins. For SOD, proteins and peroxide, the trend of causing significant changes remained in the following year, too (Appendix A), in addition to MDA, which significantly increased in 3 out of 4 varieties in 2021. In the second year, genotype VI confirmed its dominance in provoking changes by causing 25 modifications in total. In the same year, genotypes I, II, and III caused larger numbers of significant changes (21) than independently.

#### 2.2.3. Wild-Type-Induced Changes

Similar to LR treatment, PERMANOVA for parameters measured in grapevines inoculated with wild-type (WT) virus combinations (Appendix A) showed that virus treatment (R^2^ = 0.14, *p* = 0.001) and grapevine variety (R^2^ = 0.08, *p* = 0.001) had an effect on the measured parameters. Interaction between both was also statistically significant and explained 35% of the total variance (*p* = 0.001).

DAPC analysis for grapevines infected with WT isolates showed less separation in the first year only for ‘Cabernet Franc’ (Figure 5a). In addition, WT-infected plants of the ‘Tribidrag’ variety were also grouped with ‘Cabernet Franc’ plants. LD1 separated ‘Merlot’ from other varieties showing the highest loading for a set of colorimetric parameters (chroma value; avg c), but also parameters involved in photosynthesis to some minor extent. LD2 separated controls from WT-treated plants for all but ‘Cabernet Franc’ and showed the highest loading for K. Similarly, as for LR treatment, we observed a better separation of varieties and treatment in the second year with Ca as the determining factor in both LDs. The spread of plants inoculated with different WT again did not change from year to year.

‘Merlot’ was again the variety with the smallest change between the years (Figure 2b). In contrast to LR, the ‘Tribidrag’ variety showed less change, and ‘Pinot Noir’ showed more change between the years.

Common indicators of viral infection in the first year of measurement for all varieties were Fe, SOD, and proteins (Figure 3b), which were the most susceptible to changes. An indicator of viral infection was also peroxide, which was changed in 3 out of 4 varieties in the first measurement (Figure 3b). Although, as in the case with LR isolates, Fe concentration was less significant, with the duration of infection, the trend of SOD, peroxide, and proteins retained its significance. The significance of MDA, proline, chlorophyll *b,* and carotenoids appear to increase with the infection since there was a serious trend of changes in the year 2021. The trend of the ‘Cabernet Franc’ variety, which had the least changed parameters in the first year (Figure 3b), recorded the most numerous significant responses in the second. Of only five changed parameters, regardless of isolate type in ‘Cabernet Franc’ in 2020, Fe, SOD, and proteins were detected in other varieties.

#### 2.2.4. WT Genotype-Related Changes

Filtering the significantly changed parameters by introducing the effect of isolate, the trend of disturbed SOD, peroxide, and proteins in relation to control plants only increased in the year 2021 (Figure 6 and Appendix A). Microelement Fe, the most affected among all nutrients tested in the first year, was substituted by Zn in the following year (Appendix A). ‘Cabernet Franc‘was the least affected by different WT genotypes in the first but the most in the second year, approaching ‘Tribidrag’ in terms of changed parameters, especially regarding pigments concentration. Although the number of affected parameters was almost equal in both years (19 in the first and 20 in the second), with progressing time of infection, WT genotypes caused more intensive responses in host plants (Appendix A). Q isolate provoked the most numerous reactions in infected plants in the year 2021.

#### 2.2.5. Symptoms Observation

In the first season after infection (September 2020), mostly mild symptoms of GLRaV-3 infection were recorded and only sporadically on ‘Merlot’ plants infected with X, Y, Q, and Z isolates. The most frequent symptoms recorded were leaf redness, while leaf rolling was less frequent but present. Symptoms of other viruses were not present.

In the second growing season, symptoms started to develop earlier in the season, in the middle of July, for ‘Merlot’ and ‘Cabernet Franc,’ and the full symptoms expression was reached in September when ‘Pinot Noir’ also developed clear symptoms of GLRaV-3 infection. Each of the varieties developed symptoms of leaf reddening and rolling but with different intensities. Symptoms of chlorosis were more pronounced and represented in infected ‘Tribidrag’ than reddening or rolling. WT isolates in the second year also developed more symptoms with greater intensity (Appendix A). Phenology was not disturbed in any of the growing seasons.

## 3. Discussion

### 3.1. Common Host Responses to Viral Infection

The host’s response during the time of infection displayed a pattern that included changes in oxidative stress parameters rather than nutrient and metabolic processes. As SOD, hydrogen peroxide, and proteins were drastically affected regardless of the isolate type, the host, and the duration of the infection, those parameters could be used for indication of viral infection in general. Changes they caused pointed to a serious imbalance in cellular redox status, which was triggered in plants only six months post-inoculation. Incompatible virus infections, such as between GLRaV-3 and *Vitis vinifera*, the ROS production is not massive and strong enough to cause cell death; hence, the survival and spread of the virus are almost certain, and systemic plant infection is inevitable [14]. As proved by nested PCR in our experiment, the infection became systemic, and the virus continued to alter the host response to infection.

The disturbed SOD activity in our experiment confirmed the presence of stress conditions. From the drastic SOD disturbance in all varieties and the majority of genotypes (Figure 4 and Figure 6) in both years found in our study, we can assume an intensive SOD response to virus pressure. A similar finding was recorded by Cui et al. [23], but only when the grapevine was subjected to the combined effect of GLRaV-3 infection and water stress. Many authors have also found changes in SOD activity in different plant species and varieties in response to virus infection [24,25,26], and it is often activated in plants tolerant to stress [27].

Hydrogen peroxide in the first season fluctuated more, but in the second, all plants except ‘Tribidrag’ (Figure 4 and Figure 6) had increased peroxide concentration which is consistent with the results of Cui et al. [23], who also found increased hydrogen peroxide in plants grown in vitro. Díaz-Vivancos et al. [28], studying viral infection in apricots, found that with compatible interactions, an increased amount of peroxide over a longer period of time leads to increased lipid peroxidation and protein oxidation which could be the scenario with our experimental plants since peroxide continued to rise with time.

The infection of the grapevine mainly led to disturbed total protein content with no clear pattern. Certain treatments in some cultivars caused an increase and, in some, reduced the amount of proteins (Figure 4 and Figure 6), also accompanied by the disturbed nitrogen concentration in LR isolates (Figure 3a). Modulation of protein content was recorded as one of the physiological responses to biogenic stress in several studies [29,30]. Bertamini et al. [31] found a reduced amount of soluble proteins in the GLRaV-3 infected leaves, presumably connected with damage in chloroplasts or inhibition of protein synthesis, while Cui et al. [23] found an increase in proteins in infected *Vitis* plants. It is possible that with longer infection, a clearer picture of the protein status would be adopted.

MDA also showed significant and regular changes in both types of isolates, especially pronounced in longer infection, where a continuous increase in the MDA concentration was observed (Figure 3 and Figure 4f). The presenting changes in MDA, peroxide, proteins, and SOD support the presence of severe oxidative damage to host plants. The production of ROS, as well as MDA, is responsible for the partial activation of defense mechanisms and viral inhibition. To some extent, it is possible that these compounds were engaged to elevate the plant immune system to fight infection, although inhibition of ROS generation and absence of hypersensitive response is a strategy to achieve compatible infection between virus and host [32,33], most likely to be also the case in our study.

The trend of drastic carotenoids, chlorophyll *a* and *b* reduction in all isolates in 2021, as opposed to 2020 (Figure 3), agrees with the results of Bertamini et al. [17,31]. They found reduced levels of total chlorophyll and carotenoids in grapevine-infected leaves. All these pathological changes could be associated with the reduction in photosynthetic activities [21,29], as presumably would happen in our study with prolonged infection.

Enhanced or repressed values for the same measured parameters in relation to control plants and inconsistencies in their appearance could be due to sensitive mechanisms of balancing all the processes engaged in these interactions, relatively young infections, and young plants. The host’s response pattern during the time of infection has focused rather on oxidative stress parameters than nutrient and metabolic disturbances.

### 3.2. Reaction to Viral Infection Is Variety-Dependent

Most studies with similar research interests examine the physiological effects on only one variety [17,18,23,29,31,34], and if there is more than one, results are presented separately for each variety [21,35]. It is not surprising since different varieties react differently to viral treatments, as shown here by DAPC statistics when noninfected control plants were strictly variety-grouped (Figure 1 and Figure 5). The parameters analyzed were discriminating factors for some varieties, regardless of infection status. Colorimetric and photosynthetic parameters were good indicators of the difference between ‘Merlot’ and all other varieties, no matter if they were healthy or infected. Since the typical symptom of leafroll disease is leaf reddening, colorimetric parameters could not serve to discriminate infected from noninfected plants without considering the varietal factor. On the variety level, the changed concentration of chlorophyll *a* and *b*, as well as carotenoids compared to the control plants, were common to ‘Cabernet Franc’ and ‘Tribidrag’ varieties affected by different genotypes of LR and WT isolates (Appendix A). With the progression of the infection, Ca proved to be an important element in separating either the variety or viral status, especially highlighted for the ‘Merlot’ variety. This is not surprising because Ca homeostasis and Ca signaling are critical to plant immune defenses [36] which obviously also applies to viral infection. In this study, we demonstrated that grapevine genotype plays a crucial role in plant–virus interactions. This is in line with the results on grapevine fanleaf virus-infected ‘Nebbiolo’ and ‘Chardonnay’ varieties whose susceptibility to fungal disease differs for the different metabolic activities provoked by the viral attack [37].

### 3.3. LR and WT Isolates Differ in the Ability to Provoke Host Reaction

Analyses of all isolates together showed no discrimination between infected and noninfected plants, indicating that LR and WT types of isolates provoked different changes in their hosts. In LR isolates, more changes in the nutritive status and primary metabolism were observed shortly after infection (Figure 3a), but later LR effects were more manifested in the modification of stress parameters. The concentration of almost every element was changed in relation to control plants in the first season of LR-infected plants (Figure 3a). Nutrients recycled from infected leaves can be an adaptive response of plants to viral infections [20], and since plants experienced initial oxidative stress, the result was a disturbed nutrition status. On contrary to LR, changes in nutritive status for WT-infected plants were continuously focused on a few elements, among which change in Fe concentration was most pronounced at the beginning of infection (Figure 3b). The significant change in Fe could be explained by the involvement of Fe in various metalloenzymes active in biological oxidation [38], among which SOD is one most effective. With LR isolates, Zn was heavily disturbed in the first season, whereas WT infections were only in the second season. That is an important change because Zn reduces disease severity [39]. The status of nutrients is vital for the plant’s defense capability [39], but at this stage of infection, no clear pattern could be determined.

Analyzing LR and WT genotype-dependent changes, no specific response could be associated with a particular variety or parameter tested. The tendency of provoking changes in LR genotypes VI and a combination of I, II, and III genotypes has grown over the years (Appendix A). Q isolates, composed of six different viruses, expectedly caused the most numerous reactions in the second year (Appendix A).

### 3.4. The Host Response Is Time-Dependent

In our study, six months post-inoculation, drastic changes had affected all varieties, which in such a short time already statistically clearly discriminated control from the infected plants (Figure 1 and Figure 5). It is not surprising that the ELISA test mostly tested negative for the virus presence, as, in such young plants and young infection, the virus titer is below the ELISA and standard PCR detection threshold. The effect of virus amount does not seem to be necessarily positively correlated with symptom severity. Nevertheless, in our experiment, low virus titer had an influence not only on physiological parameters but also on symptom development. ‘Merlot’ pops out as a variety that reacted faster and more intensively than other varieties, confirming the status of a reliable biological indicator [40]. A similar observation was reported by Chooi et al. [41], with the ‘Merlot’ variety showing GLRaV-3 symptoms earlier in the season compared with ‘Pinot Noir’ and ‘Cabernet Sauvignon’ infected vines. With the progress of the infection, the separation between grapevine varieties was better, as ‘Cabernet Franc’ and ‘Tribidrag’ were also separated (Figure 1 and Figure 5). Progression of the infection is most obvious for ‘Cabernet Franc’ infected with WT isolates, as evident in a larger number of physiological parameters changed (Figure 3b) and symptoms development (Appendix A).

### 3.5. Symptoms Expression of GLRaV-3 Isolates in Single and Mixed Infections

Two years of symptoms observation elucidated all the differences in varietal sensibility to symptoms development by virus infection and to a type of isolate. ‘Merlot’ proved to be the most susceptible variety in terms of symptoms expression, as six months post-inoculation, only ‘Merlot’ infected with WT isolates developed symptoms of viral infection under greenhouse conditions. As expected, with infection duration, other varieties also developed typical symptoms of infection, with ‘Merlot’ and ‘Cabernet Franc’ showing symptoms earlier in the season. Compared to the optimal time frame of 2 to 4 years, which are needed for symptoms to be expressed in field condition [40,42], our study showed that by chip-bud inoculation, symptoms could be expressed much earlier, even six months post inoculation under greenhouse condition. WT isolates induced more severe symptoms in every indicator than singly GLRaV-3 isolates (Appendix A), and even ‘Tribidrag’ infected with WT isolates showed symptoms of chlorosis associated with viral disease. GLRaV-3 genotypes VI and VII, contrary to triggered physiological changes, developed minor biological symptoms in this stage of infection, which could be due to the lower titer of those variants. ‘Tribidrag’ variety, which did not develop typical symptoms of leafroll disease, but manifested numerous physiological changes, leads to a conclusion of ‘Tribidrag’ being more tolerant to symptoms development than other tested varieties.

Besides ‘Tribidrag’, which is not a standard indicator for leafroll disease and genotypes VI and VII, the antioxidant response, in general, was associated with the severity of systemic symptoms as seen for all viral isolates.

## 4. Material and Methods

### 4.1. Grapevine Plants for Grafting

A set of red grapevine indicators was included in the experiment: ‘Cabernet Franc,’ ‘Merlot,’ and ‘Pinot Noir’ obtained as certified cuttings from the Hochschule Geisenheim University, Institut für Rebenzüchtung, Germany. Croatian native cultivar ‘Tribidrag,’ known worldwide as ‘Primitivo’ or ‘Zinfandel,’ recently eradicated from all known grapevine pathogenic microorganisms at the Foundation Plant Service, University of Davis, California, USA, was also included in the experiment. The cuttings were treated with fungicides against *Botrytis cinerea* and with 2000 ppm IBA solution (Sigma Aldrich, St. Louis, MO, USA) prior to rooting in a mixture of perlite and peat (3:1). Rooting was conducted on a heating table, and cuttings were irrigated every day/every two days to prevent drying out before root formation.

Four weeks later, plants were transferred to 6 L pots filled with soil (brown soil), peat (Brill type 5), perlite (Agrilit 3, Perlite espansa), and quartz sand (Lasselberger-Knauf) in 1:1:1:1/3 ratio. Plants were watered with half-strength Hogland solution during vegetation season.

### 4.2. Viral Isolates for Grapevine Grafting

Thirteen different virus isolates for grafting were used in this study (Table 1), composed solely of GLRaV-3 (LR) or LR in combination with other wild-type viruses (WT). Five inoculums were composed of monophyletic GLRaV-3 isolates and included genomic variants from phylogenetic groups I, II, III, VI, and VII, according to Diaz-Lara et al. [3]. An additional four were prepared as mutual combinations of selected monophyletic GLRaV-3 isolates (I/II, I/III, II/III, I/II/III; Table 1). The source of LR isolates for GLRaV-3 transmission were one-year-old cuttings of ‘Cabernet Sauvignon,’ a donation by prof. Rachelle Bester from the Stellenbosch University, South Africa. WT isolates (X, Y, Q, Z; Table 1)) were composed of GLRaV-3 in combination with other viruses, most commonly present in native grapevines of the South Adriatic coast. The source for WT isolates was individual plants previously detected to be infected with particular viral composition [43]. Each type of isolate was grafted by chip-bud on five replicates plants of every variety. Untreated plants of each variety served as controls.

### 4.3. Confirmation of Virus Transmission

#### 4.3.1. Enzyme-Linked Immunosorbent Assays (ELISA)

Plants were checked for GLRaV-3 presence 3, 6, and 9 months after the inoculation. Even though the WT isolates contained other viruses. Based on the detection of GLRaV-3, we could assess whether the grafting procedure and virus transmission were successful or not. DAS-ELISA (Double antibody sandwich-ELISA) using a commercial kit (Agritest, Valenzano, Italy) was performed on fresh petioles and leaf midribs. The absorbance was recorded at 405 nm, and values 2.5 times greater than the mean absorbance value of the negative control were considered positive for virus presence.

#### 4.3.2. GLRaV-3 Detection by Reverse Transcription-Polymerase Chain Reaction (RT-PCR)

In the dormancy phase, phloem tissue served as a viral source for detection by reverse transcription-polymerase chain reaction (RT-PCR) using RNeasy Plant Mini Kit (Qiagen, Hilden, Germany). For that purpose, an improved RNA extraction procedure, as described by MacKenzie et al. [44], was applied. Reverse transcription was performed using 200 units of MMLV reverse transcriptase (Invitrogen, Waltham, MA, USA), 100 units of RNase inhibitor (Invitrogen, Waltham, MA, USA), 0.5 mM dNTPs and 2.5 µM random nonamers monomers (SigmaAldrich, St. Louis, MO USA) in the reaction mixture of 22 µL that contained 12 µL of extracted RNA. The reaction mixture was incubated for 10 min at 25 °C and 60 m at 37 °C, followed by 15 min at 70 °C. The detection of GLRaV-3 was performed as reported by Turturo et al. [45]. As indicators of RNA quality and RT-PCR effectiveness, primes for *Vitis* 18S rRNA were used. Reaction products were analyzed by agarose gel electrophoresis ascertaining the 336 bp amplicon size of GLRaV-3 and 884 bp of 18S rRNA.

In the case of negative PCR results, nested PCR for improving sensitivity and specificity was used, with the reaction template being the PCR product from the first reaction and other conditions as in regular PCR.

### 4.4. Physiological and Biochemical Measurements

#### 4.4.1. Plant Material for Analysis

For each of the four indicators, three replicate plants from each treatment were analyzed (In total: 3 replicates × 14 treatments × 4 indicator varieties = 168 samples for each parameter). Parameters of plant photosynthetic activity were measured in situ on the third completely developed leaf, which also served for measuring colorimetric characteristics. The water content and membrane permeability were analyzed on freshly harvested leaves in the period from mid-July to the beginning of August 2020 and 2021. For all other analyses, samples were collected from the end of August to the middle of September in both years. Fully developed leaves from each replicate plant were pooled and lyophilized as one sample for further analysis. All abbreviations of physiological parameters as indicators of host-plant response to viral infection are listed in Appendix A.

#### 4.4.2. Chlorophyll Analysis

The chlorophyll *a* (cla; Appendix A) and *b* were determined by using the spectrophotometric method [46], where the pigments were extracted from the lyophilized leaf with acetone (80%), and the absorbance measured at wavelengths 470 nm, 646 nm, and 663 nm. The total content of chlorophyll, total carotenoids (mg/gDW), and the ratio of total chlorophyll and carotenoids were calculated using the empirical formulas (mg/g DW).

#### 4.4.3. Photosynthesis and Stomatal Conductance

Gas exchange measurements were carried out between 10:00 a.m. and 1:00 p.m. using Li-COR 6400 (LI-Cor Inc, Lincoln, NA, USA). Calibration of the device was performed by setting the following conditions: CO_2_ concentration 400 ppm, light intensity 500 µmol m^−2^ s^−1^, 90:10 ratio of red and blue light, relative air humidity 50%, and block temperature 25 °C. Parameters measured in situ were CO_2_ assimilation rate (A), leaf transpiration intensity (E), stomatal conductance (gsw), intercellular CO_2_ concentration (Ci), and hydraulic resistance (Rh; Appendix A).

#### 4.4.4. Nutritive Status

For elements analysis, 0.5 g of lyophilized sample was subjected to dry ashing at 550 °C for 5 h, 2 mL of HCl was added and dissolved with dH20 until the final volume of 50 mL. Phosphorus concentration was determined by the method of Olsen et al. [47]. The concentration of potassium was determined on the flame photometer (Model 410, Cambridge, Sherwood, UK), and nitrogen was quantified by the Kjeldahl method (Kjeltec System 1026). Concentrations of all other elements were measured using an atomic absorption spectrometer (Spectraa 220, Varian, Palo Alto, CA, USA).

#### 4.4.5. Leaf Water Status

The relative water content was determined in leaves using the formula RWC = 100((FWDW)/(TW − DW)), which complies with the fresh mass of leaf segments (FW), the mass of rehydrated leaf segments (TW), and dry mass (DW; [48]).

#### 4.4.6. Membrane Permeability

Membrane permeability was performed according to Tarhanen et al. [49]. The differences in solution conductivity through the measurement of the percentage of electrolytes were analyzed between transversal leaf sections incubated in distilled water before and after autoclaving.

#### 4.4.7. Biochemical Indicators of Oxidative Stress and Total Protein Analysis

The activity of the SOD enzyme was determined according to the method of Beauchamp & Fridovich [50], in which superoxide anions in the presence of nitro blue tetrazolium chloride forms NBT-diformazan manifested as a color change.

Determination of malondialdehyde content, as one of the final outcomes of lipid peroxidation, was performed according to Heath & Packer [51]. The color reaction occurred due to lipid peroxide decomposition and the formation of malondialdehyde that reacts with thiobarbituric acid.

Hydrogen peroxide was measured in a reaction with titanium sulfate, forming a peroxide-titanyl precipitate, whose absorbance was measured [52].

Salicylic acid (SA) was quantified after extraction in methanol, and sample preparation was according to the method of Raskin et al. [53]. In the prepared sample, SA was quantified by monitoring fluorescence at 407 nm (excitation wavelength is 305 nm) using the method of high-performance liquid chromatography (HPLC).

The concentration of proline was determined by the spectrophotometric method [54], based on interactions of ninhydrin with this amino acid in acidic solutions.

The determination of the total soluble protein concentration was performed according to Bradford [55] and is based on protein binding to Coomassie Brilliant Blue G-250 dye, which is then spectrophotometrically detected.

#### 4.4.8. Leaf Colorimetric Characteristics

Leaf colors were analyzed using the chromometer (Konica Minolta CR-400) based on the CIELAB color system [56]. Measured values were of L (luminance), a (range from red to green), and b (range from yellow to blue). Chroma values (c) and hue angle (h) were calculated.

### 4.5. Morphometrical Measurements and Symptoms Observation

The shoot length (SL) of non-infected control and treated plants were measured as well as the number of internodes (NI). The length of internodes was calculated out of measured values (SL/NI). Flower buds were removed. During the seasonal growth, leaves were inspected for chlorosis and reddening with special attention to leaf margins for rolling, splitting, criklening, and chlorosis. Four disease symptom ranks were recognized: asymptomatic ( ), mild (*), moderate (**), and severe (***).

### 4.6. Statistical Analysis

Data used for statistical analysis are presented in Appendix A (year 2020) and Appendix A (year 2021).

Discriminant analysis of the principal components (DAPC) was used to examine the differences between the different grapevine varieties and virus treatments (LR or WT). DAPC was performed with the DAPC function of the R Adegenet v2.1.5 package [57] in R v4.1.3. To evaluate changes in varieties in both years, we used procrustes analysis. We calculated the bidirectional Procrustes correlation coefficient using the protest function in the Vegan v2.6-2 package with 9999 permutations.

In addition to individual measured parameters, we compared the overall differences between grapevine varieties and virus treatments using perMANOVA. We used the adonis 2 function in the R library vegan (v2.5-7). For pairwise comparisons between individual grapevine varieties or virus treatments, pairwise Adonis [58] was used.

## 5. Conclusions

This research detects whole spectra of plant changes triggered by the virus in four *Vinifera* varieties: changes in nutrient status, oxidative stress, and primary metabolism. A year and a half after infection, many biological parameters associated with different cellular processes were disturbed. SOD, peroxide, and proteins stand out as the most indicative parameters and could be considered cellular markers of viral infection. Apart from this, some parameters like ferrum concentration appeared to be indicative of the beginning of viral infection, while MDA, chlorophyll, carotenoids, nitrogen, and calcium concentration only after a longer period, showing time-dependent host response. Grapevine’s responses were also variety and inoculum-dependent. ‘Tribidrag’ was shown to be more adaptive to viral stress than other red varieties tested, as evidenced by the rare symptoms occurrence of leafroll disease, but disturbed oxidative balance.

## Figures and Tables

**Figure 1 ijms-24-00008-f001:**
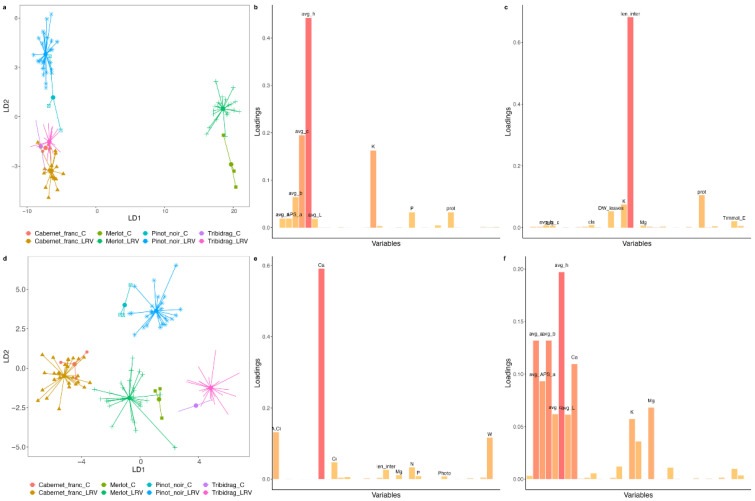
Plots generated by Discriminant Analysis of Principal Components (DAPC) from data collected for leafroll (LR) isolates in the first year of measurement in 2020 (**a**–**c**) and in 2021 (**d**–**f**). (**b**,**e**) represent variables that were important for the first discriminant function (LD1); (**c**,**f**) represent variables that were important for the second discriminant function (LD2). Abbreviations are listed in Appendix A.

**Figure 2 ijms-24-00008-f002:**
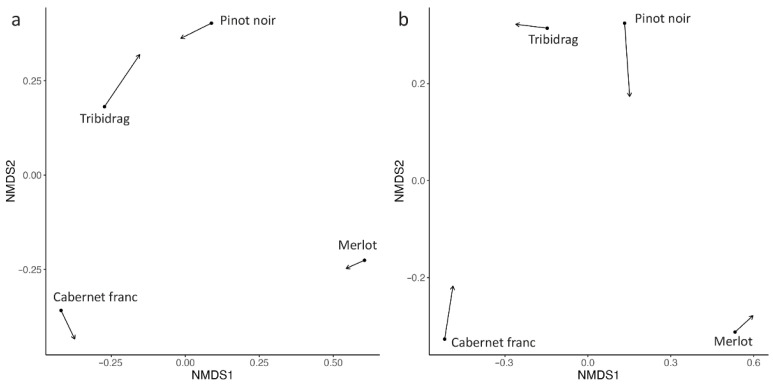
Procrustes analysis based on the NMDS plots of yearly changes (from 2020 to 2021) in grapevine parameters selected by DAPC analysis for (**a**) leafroll and (**b**) wild-type virus treatments. There was a significant correlation between the yearly changes for the leafroll treatment (R^2^ = 0.27, *p* < 0.01) and wild-type treatment (R^2^ = 0.34, *p* < 0.01).

**Figure 3 ijms-24-00008-f003:**
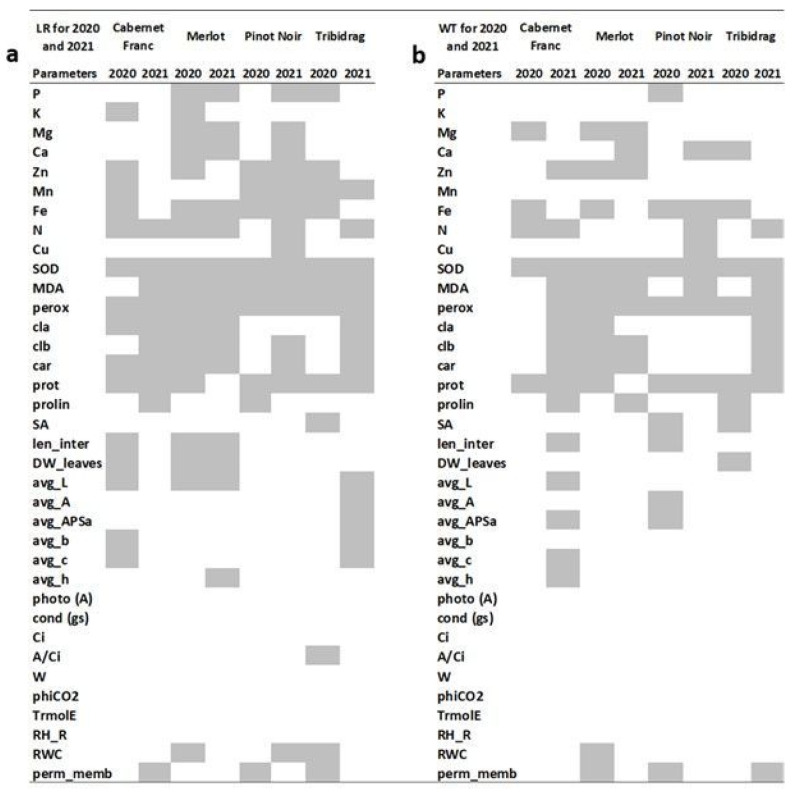
Summary of changed parameters according to ANOVA analyzed for leafroll (LR) genotypes in 2020 and 2021 (**a**) and wild-type (WT) genotypes in 2020 and 2021 (**b**) for every grapevine variety. Statistically significant (*p* < 0.05) differences are signed in grey.

**Figure 4 ijms-24-00008-f004:**
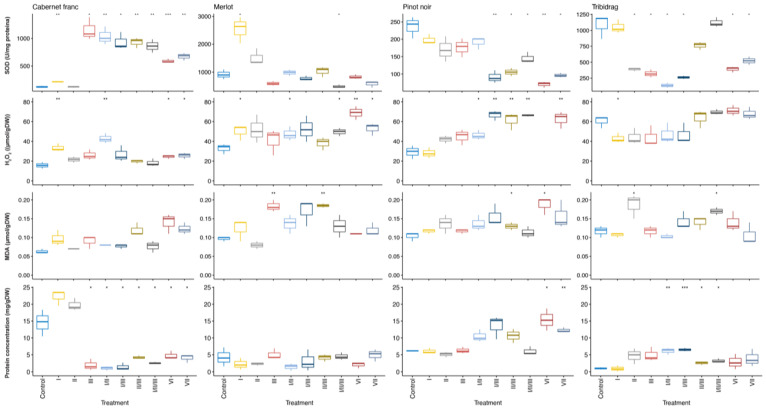
Differences in superoxide dismutase (SOD), hydrogen peroxide (H_2_O_2_), malondialdehyde (MDA), and proteins caused by leafroll (LR) genotypes in 2021 compared to control plants in ‘Cabernet Franc,’ ’Merlot, ‘Pinot Noir’ and ‘Tribidrag’ grapevine varieties as analyzed by *t*-test. Significant differences are indicated with asterisks as follows: ‘*’indicates *p* < 0.05, ‘**’ indicates *p* < 0.01 and ‘***’ indicates *p* < 0.001.

**Figure 5 ijms-24-00008-f005:**
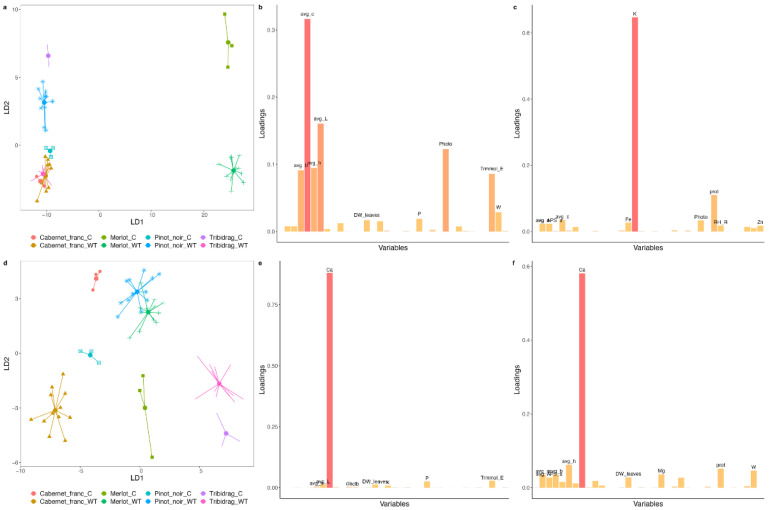
Plots generated by Discriminant Analysis of Principal Components (DAPC) from data collected for wild-type virus (WT) isolates in the first year of measurement in 2020 (**a**–**c**) and in 2021 (**d**–**f**); (**b**,**e**) represent variables that were important for the first discriminant function (LD1); (**c**,**f**) represent variables that were important for the second discriminant function (LD2). Abbreviations are listed in Appendix A.

**Figure 6 ijms-24-00008-f006:**
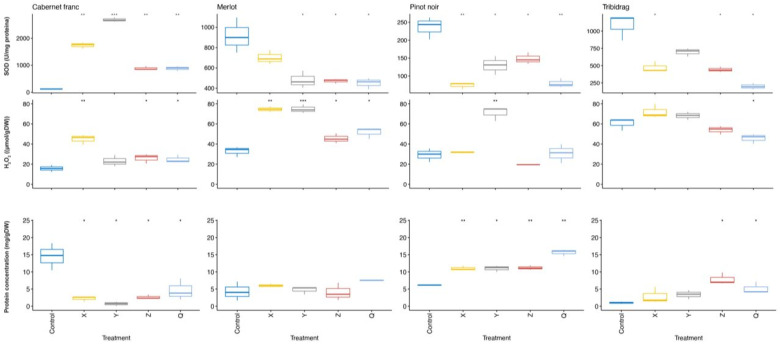
Differences in superoxide dismutase (SOD), hydrogen peroxide (H_2_O_2_), and proteins caused by wild-type (WT) genotypes in 2021 in relation to control plants in ‘Cabernet Franc,’ ‘Merlot,’ ‘Pinot Noir’ and ‘Tribidrag’ grapevine varieties as analyzed by *t*-test. Significant differences are indicated with asterisks as follows: ‘*’indicates *p* < 0.05, ‘**’ indicates *p* < 0.01 and ‘***’ indicates *p* < 0.001.

**Table 1 ijms-24-00008-t001:** Viral isolates composition and the total number of grafted grapevine plants per isolate type.

No of Isolate	Name of Isolate	Virus Composition within Isolate	Type of Isolate	GLRaV-3 Genomic Variants Within Isolate	Total Number of Grafted Plants
1	I	GLRaV-3 *	LR **	I	20
2	II	GLRaV-3	LR	II	20
3	III	GLRaV-3	LR	III	20
4	VI	GLRaV-3	LR	VI	20
5	VII	GLRaV-3	LR	VII	20
6	I, II	GLRaV-3	LR	I + II	20
7	I, III	GLRaV-3	LR	I + III	20
8	II, III	GLRaV-3	LR	II + III	20
9	I, II, III	GLRaV-3	LR	I + II + III	20
10	X	GLRaV-3, GVA, GPGV, GRSPaV	WT	I + II	20
11	Y	GLRaV-3, GVA, GLRaV-1, GPGV, GRSPaV	WT	I + II	20
12	Z	GLRaV-3, GVA	WT	I + II	20
13	Q	GLRaV-3, GVA, GLRaV-2, GFkV, GPV, GRSPaV	WT	I + II	20

* GLRaV-1,-2,-3 represent grapevine leafroll associated virus: -1,-2, and -3, GVA represents grapevine virus-A, GPGV grapevine Pinot gris virus, GRSPaV grapevine ruprestris stem pitting associated virus, GFkV represents grapevine fleck virus. ** LR represents leafroll isolates composed solely of GLRaV-3. WT presents wild-type viral isolates composed of GLRaV-3 in coinfection with other viruses, most commonly found in grapevine plants, according to Hančević et al. (2021).

## Data Availability

The data that supports the findings of this study are available in the Appendix A of this article.

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
