# Peer review of "Grapevine Leafroll-Associated Virus 3 in Single and Mixed Infections Triggers Changes in the Oxidative Balance of Four Grapevine Varieties"

_ijms, 2022, doi:10.3390/ijms24010008_

Round 1

Reviewer 1 Report

The paper presents an application of Grapevine leafroll-associated virus 3 in single and infections triggers changes in the oxidative balance of four grape-vine varieties.It is a topic of interest to the researchers in related areas but the paper needs very significant improvement before acceptance for publication.My detailed comments are as follows:

1.     The ELISA test used in the paper works well and this method is a well-established method.On the other hand ,it is not a reliable test for virus detection in plants harboring such short-term infection, and the present research is a direct application of this method without new contribution in methodological research.

2.     The authors did a good experiment and some of the phenomena presented in such as LR and WT isolates differ in the ability to provoked host reaction.The differences should be described in detail,and the authors should focus more on whether these phenomena are general characteristics and if possible,explain the reason of them.

3.     On page 8,authors said “WT isolates induced more severe symptoms in every indicator singly GLRaV-3 isolates and even Tribidrag showed some symptoms of chlorosis,presumably associated with leafroll disease.”Doesn't mixed infections make the symptoms more serious?

4.     In this paper,13 different virus isolates for grafting were used,is it necessary to use so many phylogenetic groups?

5.     The description needs to be improved,some of the paragraphs are too long.

6.     The presentation should be focused on the results. It is suggested that the authors check carefully the English writing and use standard terminologies in the technical area.

Author Response

Reviewer 1.

The paper presents an application of Grapevine leafroll-associated virus 3 in single and infections triggers changes in the oxidative balance of four grape-vine varieties.It is a topic of interest to the researchers in related areas but the paper needs very significant improvement before acceptance for publication.My detailed comments are as follows.

Authors: We kindly thank to Reviewer 1. for the time spent in reviewing the manuscript and for all suggestions made for manuscript improvement. We answered to all reviewer’s comments in this document, and also change the manuscript text to suit the reviewer recommendations.

1.The ELISA test used in the paper works well and this method is a well-established method.On the other hand ,it is not a reliable test for virus detection in plants harboring such short-term infection, and the present research is a direct application of this method without new contribution in methodological research.

Authors: We agree that ELISA test is very useful method for virus detection in older plants with long-term infections. By using this method, we tried to see if the infection could be ELISA detectible at some point. The general conclusion is that the virus titter in such short infection is not ELISA detectible. That is the reason, why we further used PCR-based methods.

Authors: We agree that methods presented in the manuscript are not new, and at any moment we did not state that we administrated, tested or developed new methods. Well known methods, from laboratory to statistical analysis were utilized to gain a new knowledge about grapevine responses to virus infection. Accordingly, many references were inserted in the manuscript, describing in details methods used.

  1. The authors did a good experiment and some of the phenomena presented in such as LR and WT isolates differ in the ability to provoked host reaction.The differences should be described in detail,and the authors should focus more on whether these phenomena are general characteristics and if possible,explain the reason of them.

Authors: We thank the reviewer for this comment. We inserted some new details about the differences between LR and WT isolates and about the common host responses as can be seen in the discussion part of manuscript. Regarding presentation of the results on differences between LR and WT, we have Figure 3 describing in detail the differences between LR and WT. In the supporting information file, we have also Figures S1 and S2 presenting in details changed parameters in relation to all genotypes tested. For the limited number of figures that could be submitted in the regular text file, we put these figures in the supplementary file. Also, since the research article is also limited by number of words, it was impossible to provide detailed changes presented in Figures 3, S1 and S2.

  1. On page 8,authors said “WT isolates induced more severe symptoms in every indicator singly GLRaV-3 isolates and even Tribidrag showed some symptoms of chlorosis,presumably associated with leafroll disease.”Doesn't mixed infections make the symptoms more serious?

Authors: Yes, mixed infections caused more serious symptoms and we change the sentence in the text to be more clear: „WT isolates induced more severe symptoms in every indicator than singly GLRaV-3 isolates (Table S4) and even Tribidrag infected with WT isolates showed symptoms of chlorosis, associated with viral disease.”

  1. In this paper,13 different virus isolates for grafting were used,is it necessary to use so many phylogenetic groups?

Authors: Viral diseases have a drastic impact on plant physiology and symptom occurrence. Single infections of the field-grown grapevine with GLRaV-3 are almost non-existing.The knowledge of the consequences of the different combinations of  GLRaV-3 infections under the same environmental conditions is limited. Even less is known about the effects of different GLRaV-3 variants in different grapevine hosts. To catalogue and characterize the effects of diverse GLRaV-3 virus infections on its hosts, we came to this concept of experiment and only by analysing as many inoculums as possible we were able to firmly gain the presented results and conclusions.

  1. The description needs to be improved,some of the paragraphs are too long.

Authors: We are aware that some of paragraphs are long but we find it important for understanding the material. Since the reviewer did not suggest which paragraph should be improved, we changed and shortened some text for which we have found to burdened the manuscript text, all indicated in the manuscript.

  1. The presentation should be focused on the results. It is suggested that the authors check carefully the English writing and use standard terminologies in the technical area.

Authors: We tried to improve the presentation of the whole manuscript in general and English language as well. All changes are highlighted in the manuscript text. We hope that introducing those changes will meet the criteria of Reviewer 1 for manuscript improvement. Since no explicit suggestions were made, we tried to detect week point of the manuscript and correct it.

Authors: We kindly thank Reviewer 1. once more and kind regards.

Reviewer 2 Report

I congratulate the authors for the analysis methods used, the mathematical models implemented and the analysis of the experimental data.

I suggest to the authors to continue the research obtained.

Author Response

We are grateful for the reviewer’s kind words. Such recognition of our work is very stimulating and we will follow your kind suggestion to continue with the research.

All the best from authors!

Reviewer 3 Report

Dear authors,
please find all the comments and suggestions within the .pdf document written as comments in comment boxes. These comments will be clearly visible in PDF Xchange viewer, or Acrobat reader, while other application might show some disturbances and I cannot guarantee for them.

However, there is a technical problem with the document I got for reviewing. The six figures (tif format) are not visible! This is probably not your problem, but the problem of MDPI system that convert the word document into PDF upon uploading the manuscript. Please, check, and send me back, so that I could check the manuscript once again!

Out of the other comments and details, I will mention here just few, while the others you will find in the comments:
- try to avoid catastrophic cliche sentences, like the first sentence in the summary;
- indicate the meaning of abbreviations in the manuscript;
- other comments, please, find in the document.

With the best regards!

Author Response

Reviewer 3.

Dear authors,
please find all the comments and suggestions within the .pdf document written as comments in comment boxes. These comments will be clearly visible in PDF Xchange viewer, or Acrobat reader, while other application might show some disturbances and I cannot guarantee for them.

However, there is a technical problem with the document I got for reviewing. The six figures (tif format) are not visible! This is probably not your problem, but the problem of MDPI system that convert the word document into PDF upon uploading the manuscript. Please, check, and send me back, so that I could check the manuscript once again!

Authors: We thank the reviewer for the engagement in manuscript improvement. We carefully read all comments and suggestions and provide answers here, as well as in the manuscript text, where all changes are highlighted.

Out of the other comments and details, I will mention here just few, while the others you will find in the comments:
- try to avoid catastrophic cliche sentences, like the first sentence in the summary;

Authors: We rearrange the abstract not to start with catastrophic cliché sentence and hopefully, we succeeded to reduce the catastrophic effect what was implied in the first version of the manuscript.

- indicate the meaning of abbreviations in the manuscript;

Authors: We indicated the meaning of abbreviations into the text, by the order of appearance. A table of inoculums names and types (WT and LR inoculums) are presented in the material and methods part of the manuscript. Table of abbreviations of all parameters are presented in Table S1 in the supplementary material.

- other comments, please, find in the document.

Authors:

  1. We renamed Tables S2 and S3 and indicated in the Supporting info file that Tables S2 and S3 are in separate documents. Tables contain 35 parameters since two parameters were measured only in one year as indicated in the Table S1: Cu was measured in 2021, and SA in 2020.
  2. New Figures are inserted into the revised version of the manuscript and we truly hope that you will be able to see them now. We also upload figures in a separate file in the submission system.
  3. All varieties names are written within apostrophes.
  4. Cabernet Franc and Pinot Noir should be written in capital letters according to OIV (International Organization of Vine and Wine: https://www.oiv.int/what-we-do/viticulture-database-report?oiv=)
  5. All other minor observations and comments made through the text file, we corrected as suggested.

With the best regards!

Authors: Thank you once more and kind regards!
